# Comment on Albantakis et al. Computing the Integrated Information of a Quantum Mechanism. *Entropy* 2023, *25*, 449

**DOI:** 10.3390/e25101436

**Published:** 2023-10-11

**Authors:** Christopher Rourk

**Affiliations:** Independent Researcher, Dallas, TX 75205, USA; crourk@jw.com

**Keywords:** integrated information theory (IIT), catecholaminergic neuron electron transport (CNET), action selection, adiabatic quantum routing, hybrid biological computer

## Abstract

Integrated information theory (IIT) is a powerful tool that provides a framework for evaluating consciousness, whether in the human brain or in other systems. In Computing the Integrated Information of a Quantum Mechanism, the authors extend IIT from digital gates to a quantum CNOT logic gate, and while they explicitly distinguish the analysis from quantum theories of consciousness, they nonetheless provide an analytical road map for extending IIT not only to other quantum mechanisms but also to hybrid computing structures like the brain. This comment provides additional information relating to an adiabatic quantum mechanical energy routing mechanism that is part of a hybrid biological computer that provides an action selection mechanism, which has been hypothesized to exist in the human brain and for which predicted evidence has been subsequently observed, and it hopes to motivate the further evaluation and extension of IIT not only to that hypothesized mechanism but also to other hybrid biological computers.

The authors are to be commended for a very thorough and thought-provoking analysis that, among numerous other things, extends IIT to a CNOT gate, a quantum computing device that is part of a hybrid computing system that necessarily utilizes digital logic devices for pre- and post-processing as well as quantum logic in the form of qubits to integrate information [1]. IIT/(Q)IIT is generally applicable to discrete/finite dimensional systems, and a typical mechanism that falls within this description is something like a CNOT gate. In practice, though, there are possible hybrid mechanisms underlying biological computing systems, such as the human brain and possibly other components of the substrate of consciousness, that are not easily captured by IIT’s formal framework. This comment will consider the further extension of that analysis to a hypothesized quantum biological adiabatic energy routing mechanism and associated hybrid computing structure in the brain.

The paper applies and extends IIT to a quantum logic gate, which is inherently a component of a gate-based quantum computer. While it is useful to evaluate integrated information in such systems, it is also important to keep in mind that extending consciousness to any physical substrate or system outside of that of the conscious observer is speculative at best, be it a digital signal processor, a gate-based quantum computer or any other substrate. For the brain—the only system that we know for certain is capable of creating the human experience of consciousness—it is more likely that the integration of information is accomplished using a hybrid biological computing system, possibly with an adiabatic quantum energy routing mechanism similar to photosynthetic energy routing rather than a gate-based digital or quantum computing process [2,3]. A simple example of a hybrid biological computing system is provided by Horsman et al., who discuss the hybrid biological computing system of a bacterial flagellum that controls a direction of rotation, where a sequence of events that cause protein phosphorylation can accumulate to cause the flagellum to change rotation. This system provides an excellent example of how the function of the computing system is often intimately associated with its physical substrate. In that hybrid biological computing system, protein phosphorylation could possibly involve a quantum biological electron tunneling process [4], but if that occurs, it would involve discrete quantum events and would not use the quantum tunneling process to directly integrate information. Instead, the information of that system appears to be integrated by the accumulation of the phosphorylated proteins at the flagellum. These collective quantum tunneling events by themselves would not satisfy the IIT integration postulate, because they can be subdivided into subsets of units that exist separately from one another [5]. As such, they would not be directly associated with the phyiscal substrate for consciousness of the bacterium. However, the entire bacterial flagellum information system cannot be subdivided without destroying the functionality of the system, and it could be characterized as having some measure of consciousness even though it is not related to the human experience of consciousness. While its output is binary at the macroscopic level—maintain rotation or reverse rotation—a large amount of information must be integrated by that system if IIT were extended to model the information integration behavior of the system at the microscopic level. IIT could also be extended to other hybrid biological computing systems, although doing so might not require (Q)IIT or other modeling of electron tunneling in proteins unless that was important to the output of the model. Likewise, applying (Q)IIT to a gate-based quantum computer or a system of digital computers that implement an artificial intelligence algorithm may need to be indexed or benchmarked relative to a hybrid biological computer such as the human brain, which is known to be capable of consciousness.

The brain would use a different hybrid computing system with different quantum mechanisms than bacterial flagellum. Rather than the widespread, discrete, quantum biological electron and proton tunneling events that may occur in proteins throughout the brain [6], a hypothesized quantum mechanical electron tunneling mechanism in catecholaminergic neurons has been proposed as part of a hybrid biological computer that could integrate information through electron tunneling in the protein ferritin to compute and implement action selection [7]. There is evidence that this catecholaminergic neuron electron transport (CNET) mechanism exists based on unusual and unexpected experimental results that were predicted by the CNET hypothesis [8,9]. It is noted that there is a great deal of skepticism regarding quantum mechanics and consciousness. For example, Stanislas Dehaene states in regard to several quantum consciousness theories that “these baroque proposals rest on no solid neurobiology or cognitive science” [10]. More recent attempts to explain consciousness as a function of “collapse of the wave function” have failed to address such skepticism [11,12]. This skepticism has caused some to conclude that any quantum consciousness theory is unlikely, in part because of the perception that “large scale coherence” would be required [13] and that empirical evidence “favors” electric fields instead of anything involving quantum mechanics [14]. This comment notes that the field of quantum biology is still quite young and that many significant discoveries have only recently been made, such as the role of chiral-induced spin selectivity (CISS) in quantum biological processes [15]. Concluding that quantum biology is unimportant to the understanding of consciousness at this point is thus seen as premature at best. In that regard, CNET is unlike and unrelated to Orch-OR and other quantum consciousness theories, not only because it is not a “quantum consciousness” theory but also because it is based on a hybrid biological computer that uses cortical cognitive processing inputs in conjunction with incoherent electron tunneling and adiabatic quantum routing to compute action initiation and selection. In that regard, it is similar to what has been observed in association with the quantum biological mechanism behind photosynthesis [16] as opposed to “large scale coherence” and gate-based quantum computing. It has also made predictions of unusual and unexpected physical phenomena that were subsequently observed and provides a neural action selection mechanism as well as a physical substrate that is capable of integrating information, which are neurological and cognitive physical functions that Orch-OR and other quantum consciousness theories do not appear to provide. CNET involves millions of cortical afferents from cortical columns and afferents from other neural structures from all over the brain that converge at the striatum, which can be clearly seen in Figure 1 below:

The dendrites of the striatal neurons that receive these cortical signals have an unusual structure that allows them to increase the pre-synaptic membrane potential of axon varicosities of very large dopamine neurons that have hundreds of thousands of such synapses, which can be clearly seen in Figure 2 below:

It has recently been shown that signals like these afferent cortical signals in Figure 1 and Figure 2 that are received at the axons and not the dendrites of large SNc dopamine neurons can cause action potentials to be generated in those neurons [20] and that those neurons release dopamine in a spatially and temporally precise manner that is capable of mediating action selection [21,22], consistent with predictions made by the CNET hypothesis. This recent unexpected and surprising discovery in the field of neurosience contradicts the earlier belief that dopamine neurons do not code movement. IIT could be extended not only to simple hybrid biological computers like bacteria but also to more complex hybrid biological computers like the hypothesized CNET action selection mechanism and these complex neurons, to evaluate its relationship to consciousness [7].

Because there are thousands of these large SNc neurons, the CNET mechanism was hypothesized as a way for those neurons to self-select within the group as a function of cortical and other afferents by routing energy to assist a small number of those neurons to reach action potential, which would also prevent seizures that could result from widespread simultaneous activation [23]. The CNET mechanism could possibly route energy between those neurons for that purpose using quantum walk routing similar to photosynthesis [24,25], which is more likely to be present in a biological system than the quantum logic gate that was considered by the authors to demonstrate how IIT could be extended to (Q)IIT. The information from these cortical afferents is integrated in the membrane potential of the axons of the large neurons that they are associated with, and each of those large neurons would also be integrated through the hypothesized CNET mechanism by virtue of strongly correlated electrons tunneling though ferritin structures. Thus, unlike “quantum consiousness” theories that are related to collapse of the wave function and panpsychism, the relationship of CNET to consiousness is through the integration of afferent neural signals, making it consistent with many consiousness theories that are based on neuroscience and cognitive science. While the quantum mechanical processes associated with the CNET electron transport physical phenomenon have not yet been modelled to a level of detail sufficient to apply them to consciousness, the macroscopic behaviour of electron transfer through ferritin arrays has been experimentally proven and could help to provide a neural action selection mechanism. The CNET mechanism in combination with the other components of the hybrid biological computer of the action selection mechanism—the cortical inputs to the striatum and the other neural processes that modulate those cortical inputs, such as the cortico-striatal–thalamic loop and the the cortico-basal ganglia-thalamo-cortical loop—satisfies the IIT integration postulate, because it is one system that cannot be subdivided into subsets of units that exist separately from one another to accomplish the function of action selection.

Because CNET is part of a hybrid biological computing mechanism for action selection, it can be modelled by extending IIT in a simplified manner that does not explicitly require solution of the many-body problem presented by tunneling electrons in the ferritin structures. One such model would be to represent the “information” at each of the large SNc neurons as I_N_(t), where there are 1 to N large SNc neurons. This information may be a time-varying scalar quantity that represents a level of triplet electrons that are generated by dopamine metabolism and stored in ferritin, which is similar in some regards to the generation of phosphorylated proteins to activate the flagellum of a bacteria, but it may also be a more complex value that is also a function of the time-varying cortical inputs at the axon synapses and how they affect the membrane potential of the axon and influence the electrons that tunnel between the neurons. That value could be approximated by I_N_(C_A−S1_(t), C_A−S2_(t) … C_A−SX_(t)), where C_A−SX_(t) is the Xth time-varying cortical input of the 1 to X cortical inputs to the axon of the neuron. IIT provides a rationale for analyzing the information content of each neuron if it were extended and adopted to model this physical mechanism instead of digital logic gates (IIT) or quantum logic gates ((Q)IIT).

In addition, each neuron is coupled to the other M = N − 1 neurons by a transfer function, Z_N−M_(t), for each of the 1 to N neurons to each of the other 1 to M neurons. This transfer function represents how electrons move between neurons though ferritin structures, which results in strongly correlated electrons that could integrate the information in the group of neurons. The physical phenomenon of strongly correlated electrons is a many-body quantum mechanical problem that is difficult to accurately model but which can be approximated at some level [25], such as by macroscopic electron behavior. The transfer function would vary as a function of time due to the loading of electrons in the ferritin structures between the neurons as well as other environmental variables. Thus, using these approximations of the information in each neuron and how that information is integrated by CNET yields:$$I(t{)}_{\mathrm{CNET}}=\int_{1}^{N}\int_{1}^{M}\int_{1}^{X}(I_{N}((C_{\mathrm{A-S1}}(t),\;C_{\mathrm{A-S2}}(t)\;\ldots \;C_{\mathrm{A-SX}}(t)),\;Z_{\mathrm{N-M}}(t))dN,dM,dX$$

The values of these parameters are presently unknown but would provide a model of the cortical information that is integrated by the physical hybrid biological computer that uses the adiabatic quantum routing of the CNET mechanism. The information associated with the activity at each neuron and the coupling to other neurons could range by orders of magnitude, such that the strongest couplings of the most active neurons would dominate the total integrated information of the hybrid biological computer, which adds an analog magnitude factor that is missing from classical IIT. Extending IIT to perform this analysis would be effectively continous and non-discrete at macroscopic scales like that of the brain.

Models of physical systems are useful, but they will always only be models. A model of a system that is functional at some level, such as CNET, is capable of being objectively tested. For example, a computer model of a nuclear power plant will not generate electric power, but it can be used to identify operational risks that can be objectively measured, such as by safety system misoperation events, and can be used to improve nuclear power plant safety, if it accurately represents the operation of those safety systems. Likewise, a model rocket will not carry humans to the moon, but it can be used to improve the safety of those actual rockets if it accurately models the behavior of critical components, such as solid rocket boosters. Mathematical equations and relationships, whether written on paper or stored in silicon memory devices, cannot create the human experience of consiousness, but they can possibly help to understand neurological diseases and disorders. It is unknown whether digital logic devices or quantum gate devices can create the human experience of consciousness. What is known is that the human brain is capable of creating the human experience of consciousness, and for a model of that type of consciousness to be useful, it must accurately reflect the physical system that is being modelled, which means understanding things like CNET or other as-yet-undiscovered mechanisms that may bear on consciousness well enough to make relevant models. We are not there yet, but Computing the Integrated Information of a Quantum Mechanism provides another useful step along that path, and it shows that IIT can possibly be modified to model the only thing that we know for certain is capable of the human experience of consciousness.

## Figures and Tables

**Figure 1 entropy-25-01436-f001:**
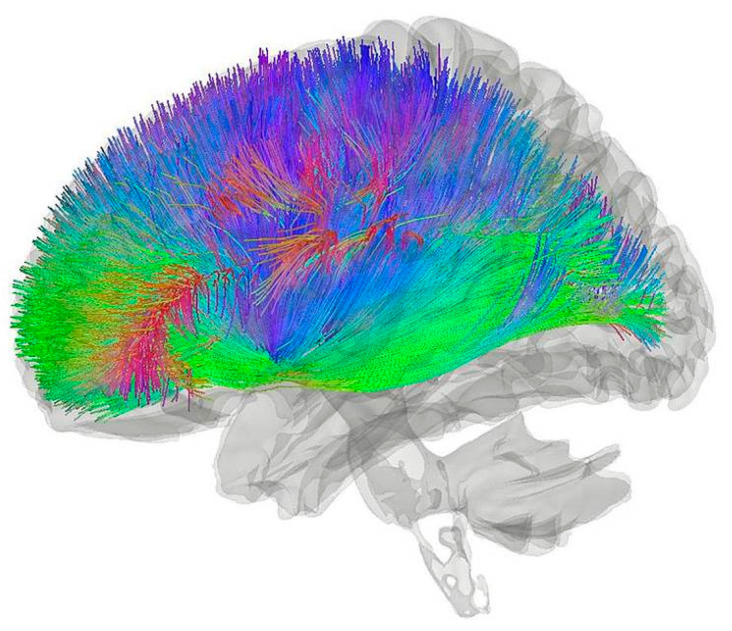
Tractography showing cortico-striatal pathway on a population-averaged template with numerous afferent signals converging at the striatum (from [17]).

**Figure 2 entropy-25-01436-f002:**
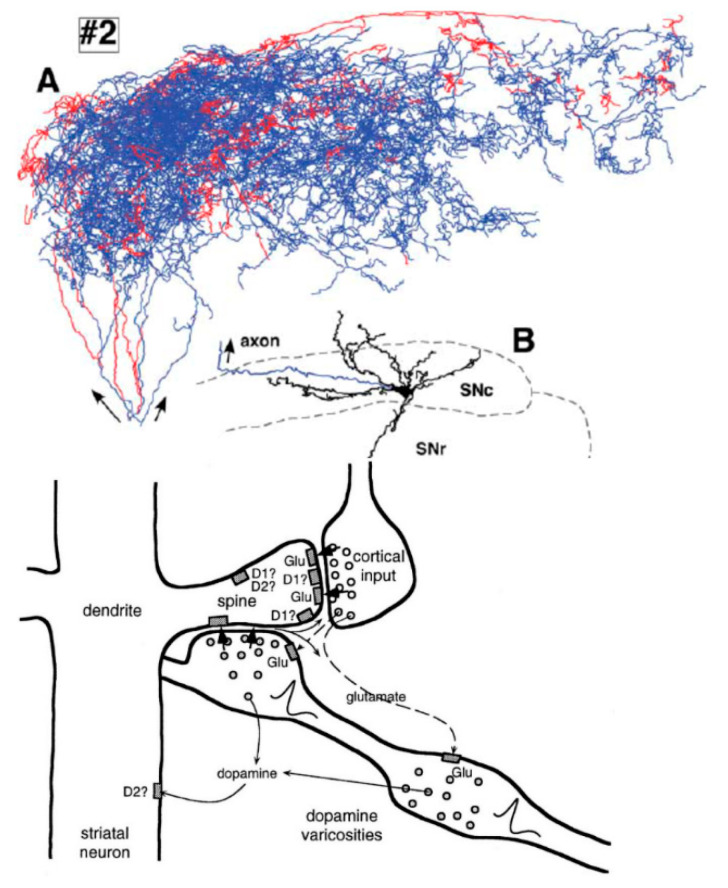
(**A**) Example of a large SNc dopamine neuron axonal arbor having hundreds of thousands of synapses (from [18]); (**B**) detailed diagram of cortical afferents and varicosities from large SNc dopamine neuron axon co-located on bouton of striatal dendrite, with neurotransmitter diffusion shown by arrows (from [19]).

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
