# Peer review of "Comment on Albantakis et al. Computing the Integrated Information of a Quantum Mechanism. *Entropy* 2023, *25*, 449"

_entropy, 2023, doi:10.3390/e25101436_

Round 1

Reviewer 1 Report

This comment addresses the recent paper by Albantakis et al. about QIIT and suggests further extension of QIIT analysis to "hybrid biological computer" in the framework of the author's CNET theory.

Unfortunately, most of the text is dedicated to the description of CNET and its supposed superiority over other theories as well as what the author calls experimental evidence supporting CNET. While I leave it to the rest of the community to trust those statements or not, I just do not understand what this comment brings to the table. The QIIT paper had a specific scope. Yes, one can think about extending it. Whether this has something to do with CNET is not clear, and the argument the author makes is not convincing, since a number of other theories/hypotheses considers quantum phenomena to be important for consciousness. Furthermore, the present comment suffers from using unclear terms (such as "hybrid biological computer"), which are not commonly accepted in the field, and lack of rigor in mathematics (which is barely used, and where it is used, some problems can be seen).

At the very least, the author should revise the Comment. It should be more succinct and focus much less on CNET. Instead, the author should clearly explain what is suggested with respect to the QIIT.

Below are several specific points. It would help if the author addressed them or removed the corresponding passages.

·  The example of a bacterial flagellum, described by the author, is not particularly enlightening. It is not clear to me what the author is trying to show. I guess it is fine to leave it in the text, but I don't quite see the point. In particular, the idea that quantum effects play any substantial role in rotation of flagella is very much a speculation (which the author acknowledges).

·  The statement "the entire bacterial flagellum information system cannot be subdivided without destroying the functionality of the system" is incorrect. The functionality depends on complex interactions of atoms within a huge number of proteins and on some chemical reactions, but nobody would seriously think that every single proton and electron comprising this system needs to be considered explicitly for an accurate description of this functionality. In fact, successful coarse-grained modeling of bacterial flagellum suggests otherwise, see, e.g., Arkhipov et al., Biophys. J., 2006 (https://doi.org/10.1529/biophysj.106.093443).

·  The text about CNET on page 2 and thereafter seems excessive. The author writes that CNET made predictions of unexpected physical phenomena that were subsequently observed, etc. I do not believe there is enough evidence to state this with such a certainty. A couple of papers by the authors himself do not mean the issue is resolved, and my impression is that most scientists in this field would be skeptical of these claims. I would therefore recommend to revise the paper in a more careful way regarding CNET and related published work.

·  The figures provided by the author are not original. Furthermore, they do not add anything new to what is stated in the text. There is large literature on the corticostriatal pathway and on dopaminergic neurons, and it is not necessary to illustrate them here. In addition, the author's points in the text do not rely on anything that is shown in these figures. Therefore, I would recommend removing them.

·  The formula provided by the author in page 4 does not make sense mathematically. What are the variables over which the integration is done, and how do their differentials enter the equation?

Reviewer 2 Report

The potential connections between Integrated Information Theory and quantum physics are deeply intriguing. This is an excellent commentary on this work, which I believe will be a valuable accompaniment to this recent publication on quantum IIT.

I believe this commentary would benefit from four minor revisions:

1. Foreground Adam Barrett's work on applying IIT to fundamental physics: https://www.frontiersin.org/articles/10.3389/fpsyg.2014.00063/full

2. More clearly mention the possibility that IIT could be valuable for helping to solve the measurement problem via a kind of quasi-Copenhagen wave-function collapse model (Chalmers): https://arxiv.org/abs/2105.02314

3. Mention the potential importance of linking quantum computing to brain function in light of recent developments in AI and the possibility of human-level intelligences (with and potentially without consciousness).

4. Briefly mention proposals in which IIT has been suggested to be valuable for quantum models of neuronal processing that potentially emulate forms of quantum computing without themselves directly involving physical quantum superposition (Safron): https://www.frontiersin.org/articles/10.3389/fncom.2022.642397/full
(Please see section the section titled "Bayesian blur problems and solutions; quasi-quantum consciousness?")

These last two points are potentially notable for extending the reach of these discussions, in that intersections between IIT and quantum computation could be immensely important for understanding and reverse-engineering the intelligence of biological systems, even if quantum computation is only of an emulated variety.

Reviewer 3 Report

In this commentary to the article, the authors discuss some of the results of the original article, as well as the nuances of applying similar and other models to describe the work of the brain.

A significant part of the analysis and reasoning of the authors of the commentary is devoted to the problems of capturing biochemical processes (and hypothetical quantum-chemical) processes using formal tools.

The reviewer agrees with the presence of many such nuances and barriers (while he strongly doubts the possibility of describing biological systems as quantum without using a phenomenological approach), but the logic of the conclusions in the commentary looks somewhat controversial.

The authors write that "Models of physical systems are useful, but they will always only be models.", further arguing that the power plant model does not provide energy, etc (sic!). The metaphor is clear, hardly anyone will argue with it, but it is very it looks like an exaggeration of the position of the authors of the article (as well as those who are engaged in modeling), since at the moment the reviewer has not met mathematicians who would be sure of the opposite.

But most of all, the tough statement raises questions - "... and for a model of that type of consciousness to be useful, it must accurately reflect the physical system that is being modelled ...".

First, absolute accuracy is not achievable in principle, as it is limited by measurement tools (even the laws of conservation of energy are described within the framework of the accuracy of an experimental study).

Secondly, it is not required. The required accuracy of the model is determined by the task. The initial meaning of modeling is the simplification of the initial task to a minimum of the processes and parameters under study (simulated), but with the preservation of the necessary and sufficient level of accuracy.

Therefore, depending on the task at hand, the same model will have different accuracy requirements, and here many examples can be given, both from physics and human models. Therefore, it is necessary to indicate what kind of accuracy is required for this model.

Therefore, in the opinion of the reviewer, this commentary is too uncompromising to modeling as a whole, contains several arguments of a rather general and controversial nature, therefore, it needs to be improved in the direction of substantive criticism of a particular model (primarily in the conclusions).
